# Predicting Species Boundaries and Assessing Undescribed Diversity in *Pneumocystis*, an Obligate Lung Symbiont

**DOI:** 10.3390/jof8080799

**Published:** 2022-07-29

**Authors:** Spenser J. Babb-Biernacki, Jacob A. Esselstyn, Vinson P. Doyle

**Affiliations:** 1Museum of Natural Science and Department of Biological Sciences, Louisiana State University, Baton Rouge, LA 70803, USA; esselstyn@lsu.edu; 2Department of Plant Pathology and Crop Physiology, Louisiana State University AgCenter, Baton Rouge, LA 70809, USA; vdoyle@agcenter.lsu.edu

**Keywords:** Taphrinomycotina, cospeciation, pathogen, species delimitation, parasite, symbiont, host switch

## Abstract

Far more biodiversity exists in Fungi than has been described, or could be described in several lifetimes, given current rates of species discovery. Although this problem is widespread taxonomically, our knowledge of animal-associated fungi is especially lacking. Fungi in the genus *Pneumocystis* are obligate inhabitants of mammal lungs, and they have been detected in a phylogenetically diverse array of species representing many major mammal lineages. The hypothesis that *Pneumocystis* cospeciate with their mammalian hosts suggests that thousands of *Pneumocystis* species may exist, potentially equal to the number of mammal species. However, only six species have been described, and the true correspondence of *Pneumocystis* diversity to host species boundaries is unclear. Here, we use molecular species delimitation to estimate the boundaries of *Pneumocystis* species sampled from 55 mammal species representing eight orders. Our results suggest that *Pneumocystis* species often colonize several closely related mammals, especially those in the same genus. Using the newly estimated ratio of fungal to host diversity, we estimate ≈4600 to 6250 *Pneumocystis* species inhabit the 6495 currently recognized extant mammal species. Additionally, we review the literature and find that only 240 (~3.7%) mammal species have been screened for *Pneumocystis*, and many detected *Pneumocystis* lineages are not represented by any genetic data. Although crude, our findings challenge the dominant perspective of strict specificity of *Pneumocystis* to their mammal hosts and highlight an abundance of undescribed diversity.

## 1. Introduction

The mycological research community faces a major taxonomic problem. Only around 144,000 species of fungi have been formally described [1], a mere 8% of the lowest estimates of true fungal diversity [2]. The deficit of defined species has numerous causes, including a lack of trained mycologists, cryptic and microscopic diversity in the kingdom, and limited exploration of obscure habitats. One poorly known fungal genus, *Pneumocystis*, exemplifies all of these challenges and as a result remains largely taxonomically uncharacterized, despite its relevance to human health and decades of research effort.

These unculturable obligate symbionts reside exclusively in mammal lungs attached to epithelial alveolar cells, and they have been detected ubiquitously in many lineages of mammals [3,4]. *Pneumocystis* is an opportunistic pathogen: it is the causal agent of fungal pneumonia in susceptible hosts, although it does not generally cause disease in immunocompetent hosts [5]. As expected for an obligate symbiont that must evade the host immune system, transmission experiments in laboratory animals reveal that *Pneumocystis* species are at least somewhat host-specific [6,7]. Because of this, it is repeatedly asserted in the literature that there may be one *Pneumocystis* species for every mammal species [8,9]. Currently, only six species of *Pneumocystis* are recognized [10,11,12,13,14] and 6495 living species of mammals are known [15,16]. However, the true degree of host specificity in this genus has not been sufficiently tested, and evidence suggests some *Pneumocystis* species may be shared by closely related hosts, such as within mammalian genera [17,18,19,20]. This evidence has led researchers to reassess assumptions of single host specificity [21,22,23]. Unfortunately, we still have little direct evidence. The lack of clarity surrounding *Pneumocystis* species boundaries and their diversity in wild mammal populations are issues that we begin to address in this work.

Despite the potential diversity within *Pneumocystis*, species discovery in the genus has been slow, exemplified by the fact that the recently named *Pneumocystis canis* collected from domestic dogs [14] is the first described species since the rabbit-inhabiting *Pneumocystis orcytolagi* in 2006 [13]. Meanwhile, several papers have been published describing genetic sequences of novel *Pneumocystis* lineages taken from diverse mammal groups, including primates, bats, and rodents, with mitochondrial genetic divergence among these lineages commonly falling between 10 and 30% [20,23,24,25]. None of the novel lineages from these publications have been described as new species.

Several factors explain the lack of taxonomic progress. Some of these are shared by all fungal taxonomy, such as the need to designate a type specimen, adhere to the Botanical Code of Nomenclature [26], and the lack of funding for this necessary but time-intensive process [27]. *Pneumocystis* presents further challenges in that these fungi exist in an obscure niche, are difficult to observe, and traditional species recognition criteria cannot easily be applied. For example, mating tests are often used to identify cryptic fungal species, but they cannot be performed without cultures. Although these traditional species-recognition criteria are difficult to apply to this genus, the issue has a ready solution: previous authors suggested that DNA sequence data should be the primary information on which we rely in this group [18,28]. With phylogenetic methods accepted as the preferred mode of species delimitation, the largest impediment to describing new *Pneumocystis* species is actually the loci that are most commonly sequenced. Two loci are often sequenced: the mitochondrial large subunit (mtLSU) and small subunit (mtSSU). These are useful targets because they are easily amplified and can be compared to existing sequences [29]. However, their utility for *Pneumocystis* systematics has been challenged [21], and at least some researchers view these two mitochondrial loci as inadequate to describe new species [30]. The recognized standards suggest that at least one nuclear locus should corroborate species distinction when formally introducing new *Pneumocystis* species [28].

Despite the limitations of single locus species delimitation, researchers have recognized the need to automate species delimitation in diverse, poorly known clades. As such, several molecular delimitation techniques can predict species boundaries based on as few as one locus, such as the Automated Barcode Gap Discovery (ABGD), Generalized Mixed Yule Coalescent (GMYC), and Poisson Tree Process (PTP) methods [31,32,33]. Such methods have been used with moderate to excellent success in many organismal groups, including fungi [34,35]. Although the dozens of existing *Pneumocystis* mtLSU and mtSSU sequences are regarded as insufficient to formally describe the unidentified fungi, we can use these data to infer where species boundaries in the genus likely lie and subsequently predict how many species might exist.

Describing species in *Pneumocystis* is a fundamental step that will foster a deeper understanding of how symbiont species boundaries relate to those of their hosts, which has important ecological and management implications for fungal parasites [36,37]. We aim to advance the important effort of documenting novel *Pneumocystis* species by estimating species boundaries and biodiversity in the genus. Using molecular species delimitation tools and available mtLSU and mtSSU sequences from a diverse set of hosts, we infer how *Pneumocystis* species boundaries correspond to those of their hosts, and we use this correspondence to predict the total number of *Pneumocystis* species. To complement these results, we have reviewed the literature and compiled a database of all mammal host species that have been screened for *Pneumocystis*, including information about screening results and available genetic data representing these lineages.

## 2. Materials and Methods

**Tree Inference—**We downloaded all publicly available mtLSU and mtSSU *Pneumocystis* sequence data representing undescribed *Pneumocystis* lineages from GenBank for which both loci had been collected from the same host, on or before 25 July 2019. Any *Pneumocystis* specimens from which only one of these loci was collected were not included, due to the necessity of multiple loci for one of our chosen species delimitation tools, Bayesian Phylogenetics and Phylogeography (BPP) [38]. In total, we compiled 127 pairs of mtLSU and mtSSU *Pneumocystis* sequences from 55 mammal host species (25 rodents, 9 bats, 14 primates, 3 carnivorans, and 4 others), including sequences from the five currently recognized species of *Pneumocystis* (Appendix A); this encompassed diversity observed at several positions in sequences from *Pneumocystis jirovecii* [29]. Although our taxon sampling is certainly incomplete, several of the chosen species delimitation methods are robust to incomplete taxon sampling and the presence of rare taxa [31,39]. We included homologous sequences from *Taphrina deformans* as an outgroup. We then aligned the mtLSU and mtSSU sequences separately using the alignment filtering tool GUIDANCE2 [40] under the MAFFT algorithm [41]. We removed any characters with a confidence score lower than 0.2, following the findings of Tan et al. [42], which suggest more aggressive filtering leads to a less accurate tree topology. We then verified and edited the alignments by eye in the program MEGA7 [43]. We concatenated these final mtLSU (404 bp including indels) and mtSSU (823 bp including indels) sequences into a single alignment. This alignment has been deposited to TreeBASE (TB2:S25117).

Using ModelFinder [44], which compares a standard set of candidate models, we found that based on the Bayesian Information Criterion (BIC) the substitution model TVM+I+G best explains these sequence data. We inferred an ultrametric tree in BEAST 2 [45] using the best-fit model. Briefly, we performed five million MCMC generations in three independent runs with a strict clock and a Yule model tree prior. We sampled parameters sampled every 1000 generations and examined convergence diagnostics by visually inspecting parameter traces and ensuring adequate effective sample sizes (ESS). After examining these diagnostics using Tracer [46], we combined the tree files of the three runs and generated a maximum clade credibility (MCCT) tree in TreeAnnotator with a 10% burn-in value. The MCCT served as the guide tree for all species delimitation analyses.

**Species Delimitation—**We chose four molecular species delimitation tools to predict species boundaries in *Pneumocystis*: ABGD, GMYC, PTP, and BPP. ABGD sorts individuals into hypothesized species such that interspecific variability of an input locus is always greater than intraspecific variability, a difference called the “barcode gap” [32,47]. On the ABGD web server (bioinfo.mnhn.fr/abi/public/abgd/abgdweb.html, accessed on 25 July 2022), we used the concatenated sequence alignment to generate a species delimitation hypothesis given a Pmin of 0.001, Pmax of 0.1, and a relative gap width (X) of 1. We report delimitation results from the initial partition, which was the same regardless of the distance calculation model (JC69 or Simple Distance).

GMYC [31] and PTP [33] both rely on input gene trees to estimate at what point branching patterns can be attributed to intra-species population genetics rather than speciation processes, but they use different algorithms. We estimated species diversity and limits with the GMYC single-threshold method using the R package “splits” [48]. The GMYC single-threshold method has been demonstrated to outperform the multi-threshold approach [48], therefore the latter method was not considered.

We estimated species limits using PTP through species.h-its.org. For this analysis, we performed MCMC sampling for 500,000 generations with one delimitation hypothesis sampled every 100 generations and a 10% burn-in value, and we report the results of the Bayesian analysis (bPTP). Although the guide tree consists of deeply divergent *Pneumocystis* clades, the entire guide tree was analyzed together in GMYC and PTP, as it has been demonstrated that including multiple related clades rather than analyzing individual subclades separately improves the performance of these methods [49].

Finally, BPP is a multi-locus method that delimits species using the multispecies coalescent model, which accounts for coalescent processes that can generate conflicting gene trees to infer a species delimitation hypothesis [38]. BPP can quantitatively compare the probability of competing species delimitation hypotheses, which allows us to compare the support for the hypotheses generated by ABGD, GMYC, and bPTP. Although BPP is typically applied to multilocus nuclear data, examples exist of accurate BPP species delimitation using mitochondrial genes alone [34].

BPP allows for collapsing hypothesized species but does not allow further division of individuals that have been assigned to the same species. Therefore, we used a maximally species-rich delimitation as our input for BPP, equivalent to the PTP delimitation with an additional split in *Pneumocystis* from the host genus *Apodemus* inferred by GMYC. Because the delimitation hypotheses are equivalent to each other with various nodes collapsed, running BPP in this fashion provides a direct comparison of the posterior probability of the three hypotheses. BPP becomes progressively more computationally intensive to the point of intractability as more hypothesized species are included, so we analyzed the three major clades (referred to broadly as *Rodents*, *Bats*, and *Primates*; Figure 1) separately. Several taxa on long branches which were unambiguously independent species with respect to the sampled taxa were removed (Figure 2). This was done because, unlike the single-locus delimitation methods, the inclusion of clusters of many species with only one individual sampled is not compatible with the BPP algorithm and prevents chain convergence [50]. We ran algorithm A00 to estimate appropriate prior values (θ and τ) due to our lack of knowledge of the population history of *Pneumocystis* [51], then ran algorithm A10 (species delimitation using a fixed guide tree) on each clade given the estimated priors. The MCMC ran for 1,000,000 generations with sampling every 5 generations and burn-in of 20,000.

***Pneumocystis*****Diversity Prediction—**To predict the number of *Pneumocystis* species in existence, we calculated the ratio of delimited *Pneumocystis* species to the 55 included host species for all four delimitation methods. Then, we multiplied these ratios by the number of described mammals to predict total *Pneumocystis* diversity.

***Pneumocystis*****Diversity Screening Database—**In order to better understand the state of *Pneumocystis* diversity research and identify important targets of future surveillance for these fungi, we performed a literature review to compile a database representing all mammal hosts screened for *Pneumocystis*. We identified all articles in which wild, domestic, or captive animals were tested for the presence of *Pneumocystis* colonization in the lungs through any method, including histology and genetic testing such as PCR amplification. Findings were cross-referenced with Chabé et al. 2012 [52], which reported all positive findings of Pneumocystis from mammals prior to 2012. Additionally, we searched GenBank for all genetic sequences associated with *Pneumocystis* prior to 14 May 2020 and identified sequences from mammal hosts that had been screened for *Pneumocystis* but not yet included in any publication. We recorded the identities of all screened mammal hosts, as well as whether they were positive or negative for *Pneumocystis* and whether they are represented by available genetic data. Finally, we summarized these findings in a database (Appendix A) and report the proportions of the entire class Mammalia as well as each mammal order that has been tested for *Pneumocystis* colonization.

## 3. Results

**Phylogeny—**For the three combined BEAST runs, ESSs exceeded 400 for all parameters, and the trends in both individual and combined trace plots were consistent with convergence. The recovered tree included *Pneumocystis* sequences from a phylogenetically diverse set of hosts spanning ≈170 My of mammalian evolution [53], providing an opportunity to compare *Pneumocystis* phylogenetic history to their mammal hosts at both deep and shallow evolutionary scales. In the MCCT inferred using BEAST, *Pneumocystis* lineages are largely segregated into three subclades, which we will refer to as “*Rodents*”, “*Bats*”, and “*Primates*”, based on the dominant (though not exclusive) host group in each clade. We recovered the topology *Rodents* + (*Primates* + *Bats*) (Figure 1), despite the closer relationship of rodents and primates within class Mammalia [54]. Additionally, *Pneumocystis* strains from several rodents appeared throughout the tree outside of the *Rodents* clade, including *Pneumocystis* from the lesser bamboo rat (*Cannomys badius*), the only sample that did not segregate into one of the three major subclades. *Pneumocystis* from one primate (*Eulemur macaco*, the black lemur) fell into the *Bats* clade rather than into *Primates*. Although marsupials are sister to placental mammals, we did not infer that *Pneumocystis* from *Marmosa murina* (Linnaeus’s mouse opossum) is sister to *Pneumocystis* from all placentals as would be expected in the case of perfect long-term cospeciation. This evolutionary discordance with mammal hosts at deep evolutionary time scales is consistent with some other work [19], but inconsistent with previous assumptions about the evolutionary history of *Pneumocystis* [55]. The relationship between the three major subclades was well supported, with posterior probability of 1.0 for the split between *Rodents* and *Primates* + *Bats* (Figure 1).

Although deep time relationships of *Pneumocystis* from placentals and marsupials did not match host relationships, we did recover instances of host-symbiont topological concordance at the tips of the tree. With the exception of *Eulemur macaco*, *Pneumocystis* from all primates were monophyletic, with two clades representing *Pneumocystis* from primarily New World and Old World primates, respectively, with some exceptions; this is consistent with previous work [24]. In the *Bats* clade, most *Pneumocystis* sequences from bats were monophyletic. Strains from all carnivorans, including representatives of cats *Felis*, dogs *Canis*, and ferrets *Mustela*, were monophyletic, with the exception of one divergent lineage from the domestic cat, *Felis catus*. Most Eurasian rodent strains also appeared together in the *Rodents* clade.

Many inferred relationships were well supported (≥0.95 posterior probability), with notable exceptions (here defined as any node with ≤0.70 posterior probability), particularly in the *Bats* clade which contains many taxa whose close relatives were not sampled, exemplified by the sparse representation of carnivorans, squirrels, lagomorphs, etc. (Figure 1). Support for relationships between *Pneumocystis* collected from poorly sampled host groups, such as marsupials (*Marmosa murina*) and pigs (*Sus scrofa domesticus*) was extremely poor, suggesting the clustering of these samples may be an artifact of long branch attraction [56,57].

**Species Delimitation—**All four delimitation methods produced different hypotheses (Figure 2), but these varied in predictable ways consistent with prior comparisons of these techniques [34,58]. ABGD delimited the fewest *Pneumocystis* species residing in the 55 sampled mammal species, 39, while PTP delimited the most *Pneumocystis* species, 53. GMYC and BPP were intermediate, estimating 47 and 48 species, respectively, (Table 1). BPP supported the species delimitation hypothesis from PTP with two exceptions: all *Pneumocystis* sequences collected from both *Apodemus* species were collapsed into one species, as were the sequences collected from domestic pigs and one strain from domestic cats (Figure 2). This unique delimitation of the *Rodents* and *Bats* clade by BPP and the retention of the PTP delimitation in the *Primates* clade were supported by high posterior probabilities (Appendix A) with no support for the ABGD and GMYC hypotheses. This implies that the highest and lowest estimates inferred by PTP and ABGD are improbable, with true *Pneumocystis* diversity likely falling between these extremes.

Most notably, no tool delimited a strict one-host-species-to-one-*Pneumocystis*-species relationship that is commonly assumed in the literature. Delimited *Pneumocystis* species were found on multiple closely related mammal species, particularly those in the same genus, throughout the tree. For example, all delimitation methods grouped sequences from *Pneumocystis carinii* and *Pneumocystis wakefieldiae* into two species found in multiple host species in the genus *Rattus*. Similar results can be seen with *Pneumocystis* in the rodent genera *Berylmys* and *Mus* and the primate genus *Callithrix*. Some hypothesized *Pneumocystis* species were even found in multiple closely related genera, such as those found in the rodents *Leopoldamys herberti* (Herbert’s giant rat) and *Maxomys surifer* (red spiny rat), as well as the primates *Callimico goeldii* (Goeldi’s monkey) and *Saguinus fuscicollis* (brown-mantled tamarin). There were also many instances of delimited *Pneumocystis* species being found in only one mammal, especially where only a single host species in a genus was sampled. To this point, the only clade in which there was no consistently inferred example of a single *Pneumocystis* species inhabiting multiple host species was the *Bats* clade, which is also the only group for which no examples of sequences from multiple species in the same genus exist.

Additionally, we inferred that some host species are inhabited by multiple *Pneumocystis* species. It is already known that laboratory rats (*Rattus norvegicus*) can harbor both *P. carinii* and *P. wakefieldiae* [12,59], so it is likely that this is the case in other mammals. However, it should be noted that some inferences of intra-host speciation, such as the GMYC and PTP hypotheses in the genus *Apodemus*, are likely over-splits that simply reflect high intraspecific diversity in certain *Pneumocystis* lineages. This was supported by BPP collapsing all *Pneumocystis* collected from rodents in the genus *Apodemus* into one species. Some of the other more unusual inferences, such as highly divergent species of *Pneumocystis* found in domestic cats (*Felis catus*) and Egyptian fruit bats (*Rousettus aegyptiacus*), are consistent with prior work on these groups [19,25].

**Prediction of Diversity—**The number of *Pneumocystis* species per host species predicted by these methods ranged from 0.71 (ABGD) to 0.96 (PTP) (Table 1). Although PTP predicted a nearly 1:1 relationship, this reflects a complex evolutionary landscape of some host genera sharing one *Pneumocystis* species and some host species harboring multiple divergent *Pneumocystis* lineages, rather than a perfect correspondence. Extrapolating these ratios to the 6495 described extant species of mammals [15,16] produced estimates of *Pneumocystis* diversity between ≈4600 and 6250 species (Table 1).

***Pneumocystis*****Diversity Screening Database—**The full database documenting all mammals that have been screened for *Pneumocystis* colonization to-date can be accessed in the Appendix A. Additionally, a living version of this database which researchers can contribute to and which we will maintain and update can be found at https://github.com/sbabbbie/Pneumocystis-database (accessed on 25 July 2022). We summarize the major findings here (Table 2 and Table 3).

Most importantly, only a small percent of all mammalian diversity has been screened for *Pneumocystis*. The entire history of *Pneumocystis* research has encompassed 240 host species, or a mere 3.70% of mammalian specific diversity (Table 2). This sampling has emphasized the order Rodentia, with 116 rodent species screened compared to the next most-sampled order, Chiroptera (bats) represented by 37 species (Table 3). This is consistent with the fact that Rodentia is the most diverse mammalian order, but 116 species still amounts to only 4% of rodent diversity. Of the 240 sampled species, 98 (41%) are not represented by any available *Pneumocystis* genetic data, which is primarily explained by species that were screened using histology alone. At higher taxonomic levels, 15 of 27 mammal orders have had at least one representative species sampled for *Pneumocystis*, while the remaining unsampled 12 orders are depauperate (171 species, 2.6% of mammal diversity) (Table 3). Sampling of mammalian diversity for *Pneumocystis* is also highly fragmented in the sense that few or only one species are typically screened per genus, in contrast to the comparatively high average species diversity of the sampled genera (Figure 3).

## 4. Discussion

Much research into *Pneumocystis* systematics is influenced by assumptions of strict host specificity and cospeciation, which our results question. With admittedly limited data, we find consistent evidence that, while strict host specificity and infection of only one mammal species is apparent, situations in which *Pneumocystis* species boundaries correspond to mammal genera are common.

By using a variety of species delimitation tools, we generated conservative and liberal estimates of global *Pneumocystis* diversity. Although these delimitation methods are crude and influenced by population factors [34,60] that are currently unknown in *Pneumocystis*, consistency among the predictions of these tools allows for more confidence in species delimitation hypotheses [61]. Notably, all methods reject a strict 1:1 relationship of *Pneumocystis* species to their mammal hosts. Wherever multiple mammals from the same genus, or closely related genera, have been sampled for *Pneumocystis*, our delimitation methods often predict that these hosts share the same symbiont species. More sampling is certainly needed, as the many instances of delimited *Pneumocystis* species corresponding to a single host might be driven by a lack of intra-generic sampling. This is supported by the fact that, although of the 142 sampled mammal genera only 18 are monotypic, 101 genera are represented by a single species that has been screened for *Pneumocystis*. The disparity of actual versus sampled intrageneric mammal diversity is likely creating a sampling artifact, inflating the inferred specificity of *Pneumocystis* in prior analyses [22]. Assumptions of strict host specificity based on experiments in which *Pneumocystis* could not be transmitted between laboratory mice and laboratory rats are now being challenged by the further sampling of the genera *Mus* and *Rattus* [6,20,23]. Thus, far the evidence for multi-host *Pneumocystis*, in this and other work [23], is strongest in rodent hosts, which may be indicative of less strict host specificity among rapidly diversifying or closely related hosts [22]. Such a question should be addressed by further research when sampling is improved. However, our results suggest that multi-host affinity may also be the case for many more *Pneumocystis* species from other poorly sampled mammal lineages, as discussed.

Inferring a tree using *Pneumocystis* sequences from 55 mammal species at once is a novel approach. Systematic research in this genus generally starts with the assumption that all *Pneumocystis* from a monophyletic host group are also monophyletic [20,24,25]. However, our results imply that the *Pneumocystis* phylogeny is discordant with its hosts at deep time scales and may be the product of pseudo-cospeciation, which involves early host switching followed by cospeciation or host switches between closely related hosts [62]. Including sequences from a broad range of hosts generates other surprising inferences, such as the possibility of a close relationship between *Pneumocystis* from domestic cats and pigs [19]. These phenomena are likely indicative of host shifts across highly divergent taxa, a possibility that has been given little consideration in the *Pneumocystis* literature. Interestingly, there is also evidence that geography can influence *Pneumocystis* phylogeny more than host relationships. For example, *Pneumocystis* collected from *Myodes glareolus* (bank vole) is highly similar to that collected from *Apodemus* rodents (Figure 2), despite the two hosts belonging to different families: Cricetidae and Muridae. Although the hosts are distantly related, the individuals from which these *Pneumocystis* sequences were collected resided in overlapping home ranges [19]. This is also seen in our prediction that one *Pneumocystis* species colonizes the primates *Callimico goeldi* and *Saguinus fuscicollis*, which are not sisters but do inhabit overlapping ranges [63]. These results suggest transmission and speciation of *Pneumocystis* might be driven by geography rather than host phylogeny in some cases, and that we cannot continue research into this fungal genus under old assumptions.

Our analysis suggests that there are thousands of undescribed species of *Pneumocystis*; if our estimates are accurate, we have described only ≈0.1% of *Pneumocystis* diversity after sampling only ≈4% of mammal diversity. Accurate delimitation of species boundaries is necessary for many areas of *Pneumocystis* research. For example, several studies investigating cospeciation dynamics between *Pneumocystis* and their hosts have been completed without describing or identifying the symbiont lineages in question [17,20,25]. However, prior work on host–parasite assemblages has demonstrated that assuming parasite lineages on different host species are also distinct species artificially inflates estimates of cospeciation [64]. Although authors acknowledge the possibility that highly similar *Pneumocystis* strains in sister hosts may be from the same species [17,20], the lack of accurate species delimitation has likely led us to overestimate the importance of cospeciation in *Pneumocystis* evolution. These studies have been designed under the assumption that cospeciation is the dominant speciation process and have rarely considered the importance of factors like geography, despite emerging evidence that it can be influential.

Understanding how parasite species boundaries correspond to their hosts is also vital to understanding infection and host transmission dynamics. Fungi represent one of the greatest reservoirs of potential emerging infectious diseases [65] and cannot be ignored. Two of the most devastating emerging diseases of wild animals are fungal: amphibian Chytridiomycosis (*Batrachochytrium dendrobatidis*) and White Nose Syndrome (*Pseudogymnoascus destructans*) of bats. The struggle to understand these previously obscure fungi has slowed progress towards treatment development, demonstrating the importance of basic research into fungal pathogens [65], while assumptions of host specificity in *Pneumocystis* have led researchers to discount the possibility of disease spillover from host jumps between distantly related taxa, instances of closely related fungal species colonizing domestic cats and pigs suggests it can occur [66]. The example of cat and pig *Pneumocystis* is especially interesting, since both animals are domesticated and have not had opportunity for close contact until fairly recently, following their domestication. This provides a counterexample to the argument that the lack of transmission between animals in zoos indicates lack of host switching potential [24]. However, further investigation into these conflicting findings and understanding host transmission potential require better-defined *Pneumocystis* species boundaries.

Our results suggest that sampling the lungs of a few mammal species in a variety of genera would take us a long way toward understanding *Pneumocystis* diversity. The difficulty of visualizing these fungi in mammal lungs with low infection levels is certainly a challenge, but with the acceptance of phylogenetic species descriptions for *Pneumocystis* [28] and the general availability of reduced representation sequencing methods, this need not hold us back. Our findings also support a relationship between host identity, albeit at a generic rather than species-specific level, and particular *Pneumocystis* species. Host of origin can therefore be considered a phenotypic character to bolster genetic identity, e.g., [67,68]. Researchers need only sequence a few nuclear loci in addition to the historically important mtLSU and mtSSU to accomplish this [28]. The abundance of unexplored habitats has been identified as another serious challenge facing the description of novel fungi (Hawksworth and Rossman 1997), and the lack of documentation of mammal lung fungi supports this. However, the challenge of documenting fungi inhabiting this niche can be overcome through collaboration with mammalogists who collect voucher specimens of wildlife, who we can encourage to collect lung tissue. Working with diverse researchers, such as botanists and zoologists, will be a key part of uncovering the Earth’s fungal biodiversity, as *Pneumocystis* has demonstrated that a host-dependent lifestyle can be a prolific driver of speciation. Although these results beg many more questions than they answer, that is exactly what we wish to point out. We hope to highlight the enormous gap between what we know and what there is to know in *Pneumocystis*. Clearly, discovering and describing this diversity will be no small task, as we estimate ≈4600 to 6250 species have yet to be described. However, it is a vital first step if we hope to unravel the evolutionary history of these fungal symbionts.

## Figures and Tables

**Figure 1 jof-08-00799-f001:**
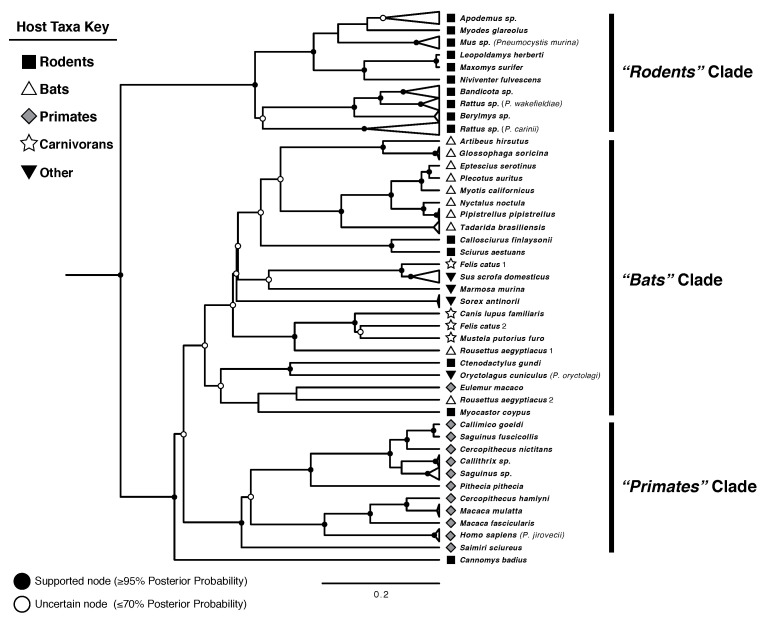
Topology of *Pneumocystis* from 55 host species recovered by BEAST analysis of concatenated mtLSU and mtSSU sequences. Sequences collected from the same host genus are collapsed where they are monophyletic for viewability, represented by triangles at branch ends. Highly supported and highly uncertain nodes are marked; unmarked nodes have posterior probabilities between 70% and 95%. Tip labels refer to the name of the host, with *Pneumocystis* species names included in brackets where sequences in the lineage were identified as belonging to a described species.

**Figure 2 jof-08-00799-f002:**
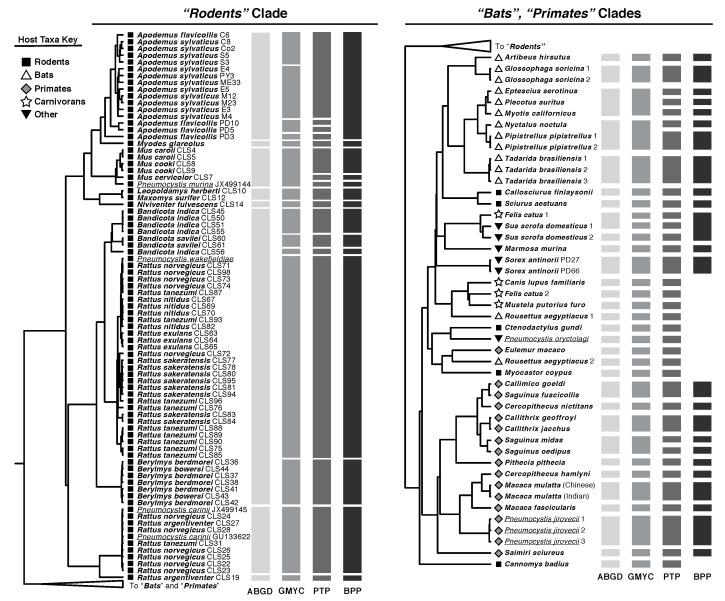
Species delimitation estimates from *Pneumocystis* in the “*Rodents*” (on **left**) and the “*Bats*” and “*Primates*” clades (on **right**) generated with four molecular delimitation methods. Node labels refer to the name of the host except where sequences were identified as belonging to a described *Pneumocystis* species (shown in underlined font). Dark bars represent the boundaries of sequences that were grouped into one species by each method.

**Figure 3 jof-08-00799-f003:**
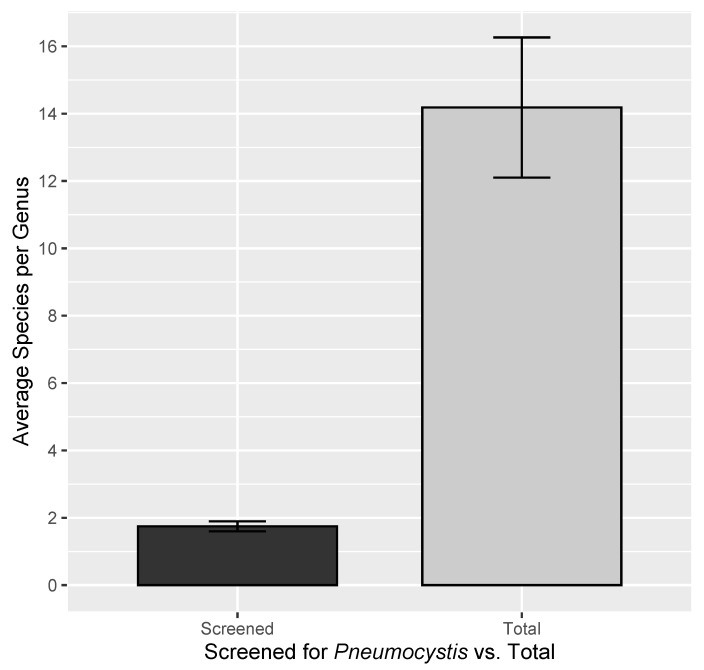
Representation of disparity between mean intrageneric diversity of *Pneumocystis* screening (**left**) and actual mean diversity of the genera sampled for *Pneumocystis* screening (**right**).

**Table 1 jof-08-00799-t001:** Inferred species delimitations and host:symbiont ratios ${}^{a}$ Although ten *Pneumocystis* samples were not included in the BPP analysis, we include them in the result here due to the confidence that they cannot be grouped into any other species. This makes the result comparable to the other three methods.

Delimitation Method	Inferred *Pneumocystis* Species	Host: Symbiont Ratio	Predicted Global *Pneumocystis* Species
**ABGD**	39	1:0.71	4606
**GMYC**	47	1:0.85	5550
**PTP**	53	1:0.96	6259
**BPP**	48 ${}^{a}$	1:0.87	5668

**Table 2 jof-08-00799-t002:** Summary of findings of literature search identifying all mammals screened for *Pneumocystis*. Number of mammal species, genera, and orders sampled for *Pneumocystis* screening. Percentages of total mammal diversity are shown as well as the proportion of species from which *Pneumocystis* genetic sequences are available.

Taxonomic Rank	Number Screened for Pneumocystis	Total in Mammalia	Percent Sampled	Pneumocystis Genetic Data Available	Percent Screened with Genetic Data	Percent Total with Genetic Data
**Mammal Species**	240	6495	3.7	129	54	1.99
**Mammal Genera**	142	1342	10.58	-	-	-
**Mammal Orders**	15	27	55.56	-	-	-

**Table 3 jof-08-00799-t003:** Number of species screened per sampled mammal order. The 12 unsampled mammal orders are not listed.

Mammal Order	Species Screened for Pneumocystis	Total Species in Order	Percent Screened
**Afrosoricida**	1	55	1.82
**Artiodactyla**	7	359	1.95
**Carnivora**	14	307	4.56
**Chiroptera**	37	1447	2.56
**Diprotodontia**	1	150	0.67
**Didelphimorphia**	2	129	1.56
**Eulipotyphlya**	17	564	3.01
**Hyracoidea**	1	6	16.67
**Lagomorpha**	3	106	2.83
**Monotremata**	1	5	20
**Perissodactyla**	1	18	5.56
**Pilosa**	1	16	6.25
**Primates**	35	516	6.78
**Rodentia**	116	2623	4.42
**Scandentia**	1	23	4.35

## Data Availability

Sequence alignments can be found at TreeBase, Project TB2:S25117. The living *Pneumocystis* diversity database can be found at https://github.com/sbabbbie/Pneumocystis-database (accessed on 25 July 2022).

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
