# Peer review of "Predicting Species Boundaries and Assessing Undescribed Diversity in Pneumocystis, an Obligate Lung Symbiont"

_jof, 2022, doi:10.3390/jof8080799_

Round 1
Reviewer 1 Report
This MS describes the analysis of Pneumocystis species boundaries using new bioinformatics tools. The observations suggest that co-evolution with their mammalian hosts is not the rule. This is an orginal and very interesting work that might become seminal in the field. However, I have major comments that should be adressed before acceptance.
Major comments :
1. The authors' must temper their argument that co-evolution has been overstated in the litterature. Indeed, their conclusion leaves a large space to co-evolution and they suggest that up to 6250 species of Pneumocystis may exist. In many places in the text (see related minor comments), the sentences should be tempered to express better that their findings is rather an adjustement of the previous bielief.
2. Most importantly, line 283, I do not agree at all with the first sentence of the discusiion : "Most research into Pneumocystis systematics rests on assumptions of strict host specificity and cospeciation":
3. Because the first clues of the relaxed host specifictiy were obtained in rodents, the authors should discuss the possibility that such less strict host species speicificity is or is not present only in small mammals. Indeed, one may hypothetisze that the transmission rules may favor jump to other species in these small species.
4. Legend fig 1: an explanation of the triangles at ends branches should be added.
Minor comments:
1. line 7: the following statement is too strong and misleading and should be tempered: "However, only six species have been described,"
2. line 34 : the following statment is also misleading and should be modified : "Because of this, it is repeatedly asserted in the literature that there may be one Pneumocystis species for every mammal species." Indeed, one species for one host species might be true in a number of cases.
3. line 197: the folowing must be clarified : "we did not recover Pneumocystis from Marmosa murina (Linnaeus’s mouse opossum) as sister to Pneumocystis from all placentals."
4. line 204: the folowing must be clarified : "Although deep time Pneumocystis relationships did not match host relationships, we...."
5. line 207: the following must be clarified : "Pneumocystis from primarily New World and Old World primates".
6. line 209: the following must be clarified, with examples : "Strains from all carnivorans were monophyletic".
7. line 214: the following must be clarified, with examples :"in the Bats clade which contains many taxa without any close relatives".
8. line 226: typo: "cats (FIGURE".
9. line 227: the folowing must be clarified : "This novel delimitation of the Rodents and Bats clade and".
10. line 231: "Leopoldomys" should read "Leopoldamys":
11. line 276: does "this" means "116 species screened" in "but this still amounts".
12. is table 3 complete, it is cut on my copy...
13. line 293: "which" should read "that".
14. line 311: the references are not correct ones : "monophyletic [20,24,25]."
Author Response
We thank the reviewer for taking the time to review and improve the submitted manuscript, as well as for their kind and constructive comments. Below is a point-by-point (numbered) response to each concern or correction, and notes on our modifications to address the concerns.
- We tempered the claims and language surrounding coevolution by addressing the reviewers further points, specified below in individual sections.
- This sentence has been modified to suggest that much Pneumocystis systematics research has been influenced by assumptions of host specificity and cospeciation, which it certainly has, rather than making the strong claim that most research in the field is beholden to these assumptions.
- Added a short discussion addressing that the strongest evidence for multi-host Pneumocystis is in rodents, and may be correlated with the rapid diversification of these species (lines 310-315). We also added a citation to our 2020 publication where we discuss this possibility at greater length.
- Clarified the text to emphasize that collapse of host genera is represented by triangles.
1. line 7: Unless we are mistaken and we have missed texts describing new Pneumocystis species, this sentence is factually accurate. Only six species have been formally described: P. carinii, P. jirovecii, P. wakefieldiae, P. oryctolagi, P. murina, and P. canis. These are the six species listed at: https://www.ncbi.nlm.nih.gov/Taxonomy/Browser/wwwtax.cgi?mode=Undef&id=4753&lvl=3&lin=f&keep=1&srchmode=1&unlock. We are happy to adjust this sentence if there is any evidence of additional species that have been formally described and accepted. Regarding the second half of the sentence, true correspondence of host-species boundaries is indeed unclear, hence our efforts in this work.
2. line 34: Again, we feel this sentence is factually accurate, as it is often repeated in the literature that there is a 1:1 relationship between host and Pneumocystis species boundaries, whether or not this is true in some cases. This sentence makes no claims one way or the other, just reports that this is a common belief in other work on Pneumocystis. We also acknowledge elsewhere in the manuscript that cases of 1:1 specificity do occur in some cases (lines 290-293).
3. line 197: We have added additional text to this sentence clarifying the meaning and implications for Pneumocystis-mammal cospeciation at a deeper evolutionary time scale (i.e. marsupials and placentals).
4. line 204: Added a clarification that this is referring to the relationship between Pneumocystis from marsupial and placental mammals.
5. line 207: Clarified that the two rodent subclades respectively represent Old and New world primates, with some exceptions, as is consistent with prior work in this group.
6. line 209: Added the common names and genera of the represented carnivorans for clarification.
7. line 214: Clarified the wording of this question and added examples of groups that have been very poorly sampled and don't have close relatives represented.
8. line 226: Thank you for catching this formatting error; fixed.
9. line 227: Clarified this sentence to emphasize that it is a "unique" rather than "novel" delimitation, as well as emphasizing that this refers to the delimitation inferred by BPP.
10. line 231: Thank you so much for catching this typo! Very helpful, appreciated. Fixed the spelling.
11. line 276: Yes, your inference is correct. Reworded the sentence for clarity of understanding.
12. Thank you, we notice this issue as well. This appears to be a problem with the LaTeX text editor that we don't know how to resolve. We will recruit the help of the formatting editors for the journal to fix this.
13. line 293: Corrected.
14. line 311: The references are correct. We use these reference to emphasize that Pneumocystis research often looks at particular host groups (e.g. primates, bats, and rodents as exemplified in these references) and assumes that all Pneumocystis from the group will be monophyletic, without including representatives from other host groups to test this assumption. We believe the references in question are good examples of this and should not be changed.
Again, we thank this reviewer immensely for their extremely beneficial commentary on this work.
Reviewer 2 Report
I found this important article well presented and despite the intricate methods used it is well written, results are remarkable, and the discussion understandable. Conclusions are according to the results and discussion. Well done.
Author Response
We thank the reviewer for this extremely kind evaluation of our manuscript. Much gratitude to you for taking the time to review this work.